**Investigation**

# Polygenic architecture of flowering time and its relationship with local environments in the grass *Brachypodium distachyon*

Nikolaos Minadakis,[1] Lars Kaderli,[1] Robert Horvath,[1] Yann Bourgeois,[2] Wenbo Xu,[1] Michael Thieme,[1] Daniel P. Woods [ID],[3,4] Anne C. Roulin [ID] [1,*]

[1] Department of Plant and Microbial Biology, University of Zürich, Zollikerstr. 107, 8008 Zürich, Switzerland
[2] DIADE, University of Montpellier, CIRAD, IRD, 34 000 Montpellier, France
[3] Department of Plant Sciences, University of California-Davis, 104 Robbins Hall, Davis, CA 95616, USA
[4] Howard Hughes Medical Institute, 4000 Jones Bridge Rd, Chevy Chase, MD 20815, USA

*Corresponding author: Email: anne.roulin@botinst.uzh.ch

Synchronizing the timing of reproduction with the environment is crucial in the wild. Among the multiple mechanisms, annual plants evolved to sense their environment, the requirement of cold-mediated vernalization is a major process that prevents individuals from flowering during winter. In many annual plants including crops, both a long and short vernalization requirement can be observed within species, resulting in so-called early-(spring) and late-(winter) flowering genotypes. Here, using the grass model *Brachypodium distachyon*, we explored the link between flowering-time-related traits (vernalization requirement and flowering time), environmental variation, and diversity at flowering-time genes by combining measurements under greenhouse and outdoor conditions. These experiments confirmed that *B. distachyon* natural accessions display large differences regarding vernalization requirements and ultimately flowering time. We underline significant, albeit quantitative effects of current environmental conditions on flowering-time-related traits. While disentangling the confounding effects of population structure on flowering-time-related traits remains challenging, population genomics analyses indicate that well-characterized flowering-time genes may contribute significantly to flowering-time variation and display signs of polygenic selection. Flowering-time genes, however, do not colocalize with genome-wide association peaks obtained with outdoor measurements, suggesting that additional genetic factors contribute to flowering-time variation in the wild. Altogether, our study fosters our understanding of the polygenic architecture of flowering time in a natural grass system and opens new avenues of research to investigate the gene-by-environment interaction at play for this trait.

Keywords: flowering time; adaptation; vernalization; *B. distachyon*; grasses; polygenic selection

## Introduction

The induction of reproduction is a critical fitness-related trait in the wild (Gaudinier and Blackman 2020), as a failure to produce offsprings during the growing season may lead to the extinction of the individual's genotype. In many annual plant species adapted to temperate climates, plantlets establish themselves in the fall and overwinter before flowering and producing seeds in more favorable spring conditions (Chouard 1960; Blackman 2017). Vernalization, the prolonged exposure to cold necessary to render plants competent to flower (Chouard 1960), is hence a key component of plant reproduction as it prevents individuals from flowering prior to winter. The adaptive potential and the genetic architecture of flowering time have been studied in an unrivaled manner in *Arabidopsis thaliana* (for review Andrés and Coupland 2012; Blümel *et al.* 2015; Takou *et al.* 2019) due to the broad geographical distribution of the species and large genomic resources developed by the community (but see Hall and Willis 2006; Monnahan and Kelly 2017; Yan *et al.* 2021 for works on other *Brassicaceae* and *Mimulus guttatus*). Vernalization is yet controlled

by different genes in different plant groups and likely evolved independently multiple times during flowering plant diversification (Ream *et al.* 2012; Bouché *et al.* 2017; Raissig and Woods 2022). Moreover, specific crop flowering-time genes (e.g. *ID1* in maize) are for instance lacking homologs in *A. thaliana* (Blümel, Dally, and Jung 2015). As a wild monocot, the model for the temperate grasses *Brachypodium distachyon* constitutes a prime system to study the evolution of flowering-time genes. In this context, we made use of the diversity panel developed for this species (Gordon *et al.* 2017, 2020; Skalska *et al.* 2020; Stritt *et al.* 2022; Minadakis *et al.* 2023) to expand our knowledge on the adaptive potential and polygenic architecture of flowering time in grasses.

Initially established as a model for bioenergy crops (International Brachypodium Initiative 2010), the grass species *B. distachyon* has more recently become a prime model for developmental biology (Woods, Bednarek, *et al.* 2017; Woods *et al.* 2019; Nunes *et al.* 2020; Hasterok *et al.* 2022; Raissig and Woods 2022; Zhang *et al.* 2022; Nunes *et al.* 2023), evolutionary genomics (Eichten *et al.* 2016; Bourgeois *et al.* 2018; Stritt *et al.* 2018, 2020; Gordon *et al.* 2020, 2017), and molecular ecology (Del'Acqua *et al.*

2014; Wilson *et al.* 2019; Skalska *et al.* 2020; Stritt *et al.* 2022; Minadakis *et al.* 2023). In addition to a chromosome-level genome assembly of 272 Mb (International Brachypodium Initiative 2010), a diversity panel composed of 332 accessions spanning from Spain to Iraq has been sequenced (Gordon *et al.* 2017, 2020; Skalska *et al.* 2020; Stritt *et al.* 2022; Minadakis *et al.* 2023), opening new avenues of research in this system (Minadakis *et al.* 2023). We previously showed that *B. distachyon* accessions cluster into three main genetic lineages (A, B, and C), which further divide into five main genetic clades: the ancestral C clade in Italy and Balkans, the B_West clade in Spain and France, the B_East clade spanning from Turkey to Caucasus and Iraq, the A_Italia clade in Italy, as well as the A_East clade in Turkey and Greece (Stritt *et al.* 2022; Minadakis *et al.* 2023). These natural accessions are found in diverse habitats (Bourgeois *et al.* 2018; Minadakis *et al.* 2023) making *B. distachyon* an ideal model to investigate how genetic and environmental factors interact to shape traits.

*B. distachyon* accessions display large phenotypic variation with regard to flowering time (e.g. Ream *et al.* 2014; Gordon *et al.* 2017; Sharma *et al.* 2017; Woods *et al.* 2019), with some accessions requiring little to no vernalization to flower rapidly (early-flowering accessions) in certain photoperiods, while other accessions require a few weeks to several months of vernalization in order to flower (late-flowering accessions; Ream *et al.* 2014; Gordon *et al.* 2017; Woods *et al.* 2019). These flowering differences have been described as potentially adaptive and responsible for population distribution according to climate variation (Gordon *et al.* 2017; Woods *et al.* 2019; Skalska *et al.* 2020). However, the extent to which variation in vernalization requirement and ultimately flowering time correlates with local environmental conditions has yet to be formally tested in this species. For instance, whether late-flowering genotypes, which require long vernalization treatments, have been selected to complete their life cycle at a slower rate to overcome harsher or longer winter, as observed in Swedish populations of *A. thaliana* (Ågren *et al.* 2017), remains an open question.

In this study, we decompose flowering time into four flowering-time-related traits: the minimum threshold duration (MTD) of vernalization necessary to be permissive for flowering, the number of days to flower when MTD is received, the saturating threshold duration of vernalization above which no further reduction in flowering time is gained, and the fastest time for a given accession to flower after vernalization saturation. We combined measurements under greenhouse and outdoor conditions and asked (i) Do flowering-time-related traits correlate with environmental variables and show signs of local adaptation? (ii) What is the respective contribution of flowering-time genes to flowering-time-related trait variation? And (iii) Are known flowering-time genes contributing to flowering-time variation in the wild?

## Materials and methods
### Biological materials and genomic resources
The *Brachypodium distachyon* diversity panel is composed of 332 natural accessions for which whole-genome sequencing data are publicly available (Gordon *et al.* 2020, 2017; Skalska *et al.* 2020; Stritt *et al.* 2022). For the flowering-time experiment, we selected a subset of 61 accessions representing all five genetic clades as first described by Stritt *et al.* (2022) and occurring along latitudinal gradients. Maps were drawn QGIS (version 3.16).

We also made use of the raw vcf produced by Minadakis *et al.* (2023) for the entire diversity panel. We used vcftools (Danecek *et al.* 2011) to apply the following filtering criteria: –max-alleles 2 –max-missing-count 200 –minQ 20. We further filtered

heterozygous SNPs as those have been shown to result from duplicated sequences and be mostly artifactual in selfing species (Stritt *et al.* 2022). All the analyses were performed on version3 of the *B. distachyon* genome (https://phytozome-next.jgi.doe.gov).

### Flowering-time-related trait measurements
We performed an experiment in controlled conditions from October 2021 until May 2022 in order to test the flowering phenology of the five genetic clades of *Brachypodium distachyon*. Twelve accessions per genetic clade were selected, in addition to the reference accession Bd21. All accessions were treated with five vernalization periods spanning from 2 to 10 weeks, with three replicates per treatment and per accession. Seeds were stratified for at least two weeks before the experiment, and then sowed in pots that were placed in greenhouse conditions (16 h day at 20 °C and 8 h dark at 18 °C with a light intensity of 200 μmol/m$^2$/s). We distributed the replicates randomly across trays to minimize bias due to position effects. Three weeks after germination, the plants were transferred to a cooling chamber (constant temperature at 4 °C, 8 h light 80 μmol/m$^2$/s, and 16 h dark) for 2, 4, 6, 8, and 10 weeks. At the end of the vernalization treatment, plants were moved back to the greenhouse.

Flowering time was measured as the number of days after return to greenhouse to the first day of spike emergence which is consistent with stage 50 of the Zadoks scale that was used in Ream *et al.* (2014). Measurements were taken every two days until the end of the experiment in May. During the experiments, trays were permuted on the table to further limit position effects.

We then extracted from this experiment four flowering-time-related traits: (i) the MTD of vernalization necessary to be permissive for flowering, (ii) the number of days to flower when the minimum threshold is received, (iii) the saturating threshold duration of vernalization above which no further reduction in flowering time is gained, and (iv) the fastest time for a given accession to flower from the day plants were moved to the vernalization chamber, i.e. the number of days to flower when vernalization saturation is received. This latter trait is classically used as a measure of flowering time in *B. distachyon* studies. Results were plotted in R version 4.0.2 (R Core Team 2018) with the package ggplot2 (Wickham 2016).

### Extraction of current and past bioclimatic variables
Raster maps for current monthly solar radiation and altitude were retrieved from worldclim (https://www.worldclim.org). In addition, raster maps for monthly Global Aridity Index (GAI) were obtained from https://cgiarcsi.community/data/global-aridity-and-pet-database/. Bioclimatic variables were then extracted using the R packages raster (v.3.5-2; Hijmans and van Etten 2012) and rgdal (v.1.5-27; Keitt *et al.* 2010) for each of the 332 accessions. For solar radiation and GAI, data were also average over spring months (April to June).

For paleo-bioclimatic variables, we used the niche suitability projections for the last glacial maximum (LGM) computed by Minadakis *et al.* (2023). For each genetic clade, we extracted the coordinates of a set of 200 random points per clade in highly suitable habitats (>0.85) with the raster package function rasterToPoints. We retrieved raster maps for LGM from https://www.worldclim.org/data/v1.4/paleo1.4.html and extracted the 19 paleo-bioclimatic variables for the corresponding sites as described above.

## Redundancy analysis

We extracted the 19 classical bioclimatic variables as well as elevation, aridity, and solar radiation in spring for the 56 accessions for which flowering-time-related traits were measured. We performed a principal component analysis (PCA) with the R base function prcomp using the resulting 22 variables.

Association between flowering-time-related traits, bioclimatic variables, and genetic cluster of origin were tested with redundancy analysis (RDA) using the R package Vegan 2.6-4 (Oksanen *et al.* 2022) following Capblancq and Forester (2023). RDA computes axes that are linear combinations of the explanatory variables. In other words, this method seeks, in successive order, a series of linear combinations of the explanatory variables that best explain the variation of the response matrix (Borcard *et al.* 2018). It is therefore especially suited when explanatory variables are correlated.

To identify which variables influence flowering-time-related traits, we opted for a forward selection approach (Capblancq and Forester 2023) and first ran empty models where the respective flowering-time-related traits were treated independently as a response variable and explained by a fixed intercept alone. We then ran additional models where each flowering-time-related traits were treated independently as response variables while the cluster of origin, the 22 environmental variables mentioned above, or both the cluster of origin and the 22 environmental variables were successively entered as explanatory variables. These full models were compared to their respective empty models with the Vegan ordiR2step function (Capblancq and Forester 2023) and the following parameters: permutations = 1000, R2scope = T, and Pin = 0.01.

We also tested for associations between flowering time and SNPs in known flowering-time genes using RDA. To do so, we first extracted and converted the SNPs located in flowering-time genes as a dataframe with vcftools—bed –extract-FORMAT-info GT, where the bed file contained the position of flowering-time genes. We first fitted simple models where each flowering-time-related trait was treated independently as a response variable and all SNPs individually as explanatory variables. Here again, we compared those models with their respective empty models with the Vegan ordiR2step function. For data visualization, the SNPs were pruned to remove SNPs in perfect linkage disequilibrium (LD) within each gene with plink (Purcell *et al.* 2007) with the following parameters –indep-pairwise 50 5 1. We kept one focal SNP per gene displaying the strongest association with one of the flowering-time-related traits for further analysis. In a second step, we fitted models where all focal SNPs were treated as explanatory variables together. We eventually fitted a full model where the cluster of origin together with all focal SNPs and bioclimatic variables associated with the given flowering-time-related trait were treated as explanatory variables. All the plots were produced with the R package ggplot2 (Wickham 2016).

## $F_{ST}$ calculation

Single SNP $F_{ST}$ between accessions of the A and B lineages were calculated with vcftools (Danecek *et al.* 2011). To account for shifts in the observed $F_{ST}$ values caused by the population structure of *B. distachyon*, the expected $F_{ST}$ distribution under neutral evolution was also estimated using forward simulations run in SLiM version 3.4 (Haller and Messer 2019). The population structure, effective population sizes, and time of divergences between lineages of *B. distachyon* during its evolution were modeled in SLiM based on the results of Minadakis *et al.* (2023). No migration between the

different populations in the simulations was allowed, as a lack of interbreeding between the distinct *B. distachyon* clades was reported (Stritt *et al.* 2022). The simulation was run 100 times and single SNP $F_{ST}$ was calculated for each simulation to generate the expected $F_{ST}$ distribution under neutrality.

## LD analyses

To plot LD decay, we first thinned the vcf with vcftools—thin 20,000 to keep one SNP every 20 kilobases (kb). Intrachromosomal LD ($r^2$) was calculated with vcftools –geno-r2. We repeated this step by further filtering the vcf for a minimum allele frequency of 0.05 with vcftools –maf 0.05. For both outputs, we visualized LD decay by plotting $r^2$ and its 95% confidence interval (CI) as a function of the physical distance between SNPs with the R package ggplot2. LD between focal SNPs in flowering-time genes was calculated separately with vcftools –geno-r2 and added to the LD decay plot for comparison. LD plots for focal SNPs were produced with the R package gaston (Perdry and Dandine-Roulland 2020).

Inter-chromosomal LD ($r^2$) was calculated with vcftools –interchrom-geno-r2 for a subset of 100,000 loci selected randomly in the genome. We then re-sampled different combinations of loci 50,000 times and calculated the mean LD each time. This allowed us to further calculate the CI around the mean and compare our real data to this distribution.

## Scans of selection

The $X^T X$ analysis was performed with BayPass v2.3 using the five genetic clades or three genetic lineages as populations (Gautier 2015). We generated the input file by using vcftools –count to calculate the allele frequency of each SNP present in our vcf (no filtering on minor allele frequency). We then ran Baypass on our actual dataset with the following parameters: –pilotlength 500 –npilot 15 –burnin 2500 –nthreads 6. To calibrate the $X^T X$ and define a threshold of significance for differentiated SNPs, we then used the simulate.baypass function from baypass_utils (Gautier 2015) to generate a pseudo-observed dataset (POD) of 100,000 loci based on the covariance matrix computed with our real dataset. We then ran Baypass on the POD with the parameters described above. We used the 0.99 quantile of the $X^T X$ calculated for the POD as a threshold of significance for the real dataset. Integrated haplotype scores (iHS) were also computed for accessions of the A and B lineages separately with the R package Rehh 3.1 (Gautier and Vitalis 2012).

## Functional effect and age estimates of variants

The functional effect of variants was annotated using SnpEff version 5.0e (Cingolani *et al.* 2012) using default parameters and the provided database for *Brachypodium distachyon*.

The age of each single SNP was computed with GEVA (Albers and McVean 2020) to estimate the average SNP age for each annotated gene (https://phytozome-next.jgi.doe.gov) in the derived A and B genetic lineages as well as in the five genetic clades. All private SNPs to the combined A and B lineages were polarized using the ancestral C lineage using custom R scripts. GEVA was run on the five main scaffolds (corresponding to the five chromosomes) using the genetic map produced by Huo *et al.* (2011) and the polarized SNP dataset.

## Genome-wide association analysis

We further used the flowering-time measurements performed outdoor in Zürich in 2017 (Stritt *et al.* 2022). We extracted the 19 classical bioclimatic variables as well as altitude, aridity, and solar radiation for the 332 accessions as well as a site in Zürich

(lat = 43.73693085, lon = 3.69295907) and performed a PCA as described above. Linear-mixed model analyses were performed as described above for the greenhouse experiment.

To identify loci associated with flowering-time variation, we performed a genome-wide association analysis (GWA) with GEMMA (Zhou and Stephens 2012). We corrected for population structure by creating a centered relatedness matrix with the option –gk 1. Association tests were performed using the option –maf 0.05 to exclude SNPs with minor allele frequency with values less than 5%. Regions were considered significantly associated if displaying at least four markers above FDR threshold in 8 kb windows (overlap of 4 kb). Upset plots were drawn with the R package UpSetR (Conway *et al.* 2017).

## Overlaps with environmental association analyses

We made use of the environmental association analyses performed by Minadakis *et al.* (2023) to assess whether flowering-time genes and the candidate gene identified by the GWAs were associated with current environmental variables. Upset plots were drawn in R with the R package UpSetR (Conway *et al.* 2017).

## Gene duplication

We checked for potential gene duplication with detettore (https://github.com/cstritt/detettore), a program developed to study structural variation based on short-read sequences (Stritt *et al.* 2021). We also calculated the proportion of heterozygous sites over the 22 flowering-time genes and 332 accessions, using the raw vcf produced by Minadakis *et al.* (2023) not filtered for heterozygous SNPs but with the following criteria: –max-alleles 2 –max-missing-count 200 –minQ.

## Results

### Flowering-time measurements under greenhouse conditions

We selected 61 accessions (Fig. 1a) from the *B. distachyon* diversity panel (Minadakis *et al.* 2023) for our flowering-time experiment. Those accessions were chosen to represent all five genetic clades and occur, when possible, along latitudinal gradients. Briefly, we submitted plants to five vernalization treatments (2, 4, 6, 8, or 10 weeks at 4 °C) with three replicates each and measured how long plants took to flower after the return to warm conditions (Supplementary Fig. 1 and Supplementary Table 1 for the raw data). For five out of the 61 accessions (Veg12 from A_East; Ren4, Lb1, Lb23, and Cm7 from A_Italia), none of the replicates flowered by the end of the experiment (Fig. 1a and Supplementary Fig. 1) despite normal growth. All subsequent analyses were hence performed on the 56 accessions for which flowering-time data were collected.

However, *B. distachyon* accessions span a large range of habitats (Minadakis *et al.* 2023). While some accessions of the diversity panel can experience up to −11 °C as a minimum temperature during the coldest months (Bio6), the mean temperature during the coldest quarter (Bio11) is above 4 °C for a large number of accessions (Supplementary Table 2 and Supplementary Fig. 2). As such, not all accessions may experience long vernalization at 4 °C in the wild. We thus extracted four traits linked to vernalization requirements and flowering time (Supplementary Table 3) for further analyses: (i) the MTD of vernalization necessary to be permissive for flowering (hereafter MTD), (ii) the number of days to flower when MTD is received (hereafter days to flower after MTD), (iii) the saturating threshold duration of vernalization

above which no further reduction in flowering time is gained (hereafter vernalization saturation), and (iv) the fastest time for a given accession to flower from the day plants were moved to the vernalization chamber, i.e. the number of days to flower when vernalization saturation is received (hereafter days to flower after vernalization saturation). Note that all plants germinated within 5 days and germination time was not included for further analyses.

Flowering-related traits show significant albeit moderate correlation among each other (Fig. 1b) and, with the exception of days to flower after MTD, a partitioning of the phenotypes per genetic clade (Fig. 1c and Supplementary Table 4 for *P*-values). Accessions from the C, B_East and B_West clades show a significantly reduced MTD compared to accessions from the A_East and A_Italia clades. Accessions from the B_East and B_West also tend to show a shorter saturating threshold duration of vernalization compared to accessions from the three other clades (Fig. 1c and Supplementary Table 4). Eventually, accessions from the C, B_East and B_West clades flower significantly faster when vernalization is saturated (Fig. 1c and Supplementary Table 4). Altogether, these results reflect that accessions from the A lineage display overall a longer life cycle (late-flowering genotypes) than the ones from the B and C lineages (early-flowering genotypes) due to longer vernalization time requirements but longer time to flower after the return to warm conditions (Supplementary Fig. 1).

### Association between flowering-time-related traits and bioclimatic variables

To test whether flowering-time-related traits correlate with environmental variation, we extracted the 19 classical worldclim bioclimatic variables (Bio1 to Bio19) as well as solar radiation in spring, GAI in spring, and altitude for each locality and performed a PCA with the resulting 22 variables (Supplementary Table 3). The first two axes of the PCA explained together about 58% of the variation among our samples, indicating that our set of selected accessions occurs in different environmental conditions.

We have previously shown that the five *B. distachyon* occur in different ecological niches (Minadakis *et al.* 2023) leading to a confounding effect of population structure and environmental variation. As mentioned above, we also observed in the current study some partitioning of the phenotypes per genetic clade for the flowering-time-related traits we investigated. Eventually, most bioclimatic variables display correlation among each other in our system (Minadakis *et al.* 2023; Supplementary Fig. 3). We therefore used a RDA to test for associations between flowering-time-related traits and bioclimatic variables, as this approach typically accounts for confounding factors by maximizing the genetic variance explained by a set of environmental predictors.

Specifically, to model the relationships among flowering-time traits, bioclimatic variable, and population structure, we fitted models where flowering-time-related traits were independently entered as response variable while the cluster of origin (model = cluster alone), the 22 environmental variables (model = all bioclimatic variables), or both the cluster of origin and the 22 bioclimatic variables (model = cluster + all bioclimatic variables) were successively entered as explanatory variables. We compared those full models to empty models where the respective flowering-time-related trait was explained by a fixed intercept alone (Capblancq and Forester 2023).

Consistent with the phenotype distributions (Fig. 1c), the genetic cluster of origin alone contributes to MTD ($R^2 = 0.25$), days to flower after vernalization saturation ($R^2 = 0.44$), and vernalization

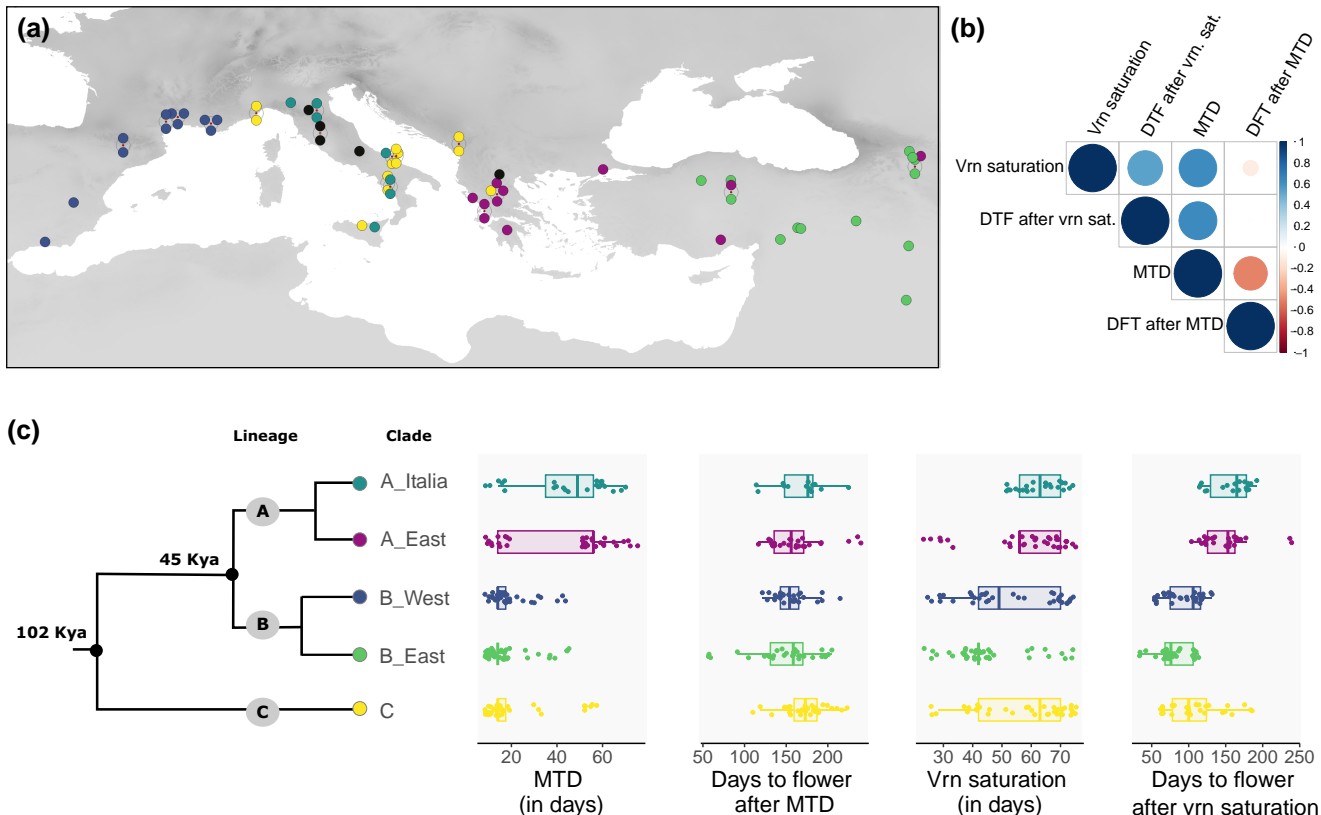

**Fig. 1.** Geographical origins of the samples used for greenhouse experiment and flowering-time-related trait variation. a) Map displaying the location of a given accession as well as their genetic clade of origin (C in yellow, A_East in magenta; A_Italia in turquoise, B_East in green, and B_West in dark blue). Accessions that did not flower by the end of the experiment are depicted in black. b) Correlogram of the four flowering-time-related traits among accessions (Pearson correlation assuming linear relationship). DTF: days to flower; MTD: minimum threshold duration of vernalization. c) Schematic phylogeny and distribution of the four flowering-time-related traits.

saturation ($R^2 = 0.1$) but not to days to flower after MTD (Table 1). We found that a large part of the variance in flowering-time-related traits is explained by bioclimatic variables (precipitation levels and temperature) linked to warm months but only marginally by elevation or variables linked to cold months (e.g. bio19; Table 1) when the models did not include the cluster of origin. Eventually, environmental (bioclimatic) variables still explained a significant part of the variance in flowering-time-related traits (Table 1) even in the full model including also the genetic clade of origin (model cluster + all bioclimatic variables). This indicates that flowering-time-related traits, when measured under greenhouse conditions, are significantly shaped by the current environmental conditions we tested in *B. distachyon* and are locally adapted.

Note that variable selection optimizes the variance explained by explanatory variables but does not necessarily identify the ecological drivers of phenotypic variation. In our system, where bioclimatic variables are highly correlated with each other and sometimes with the genetic clade of origin (Supplementary Fig. 3), while one variable may maximize the variance, another one might be the mechanistic driver of variation (Capblancq and Forester 2023). As such, it remains difficult to disentangle the effect of the cluster of origin from the effect of the environment for MTD and days to flower after vernalization saturation (Table 1) and to estimate the actual part of the variance explained by the environment for those two traits.

It is yet clear that accessions from more arid environments tend to flower faster once vernalization saturation has been reached. Indeed, Kendall rank correlations between flowering time and aridity in spring or annual precipitation levels (bio12) for instance show significant (P-value = 7.143e-06 and 4.982e-11, respectively) but only partial correlations (Tau = −0.25 and 0.36, respectively), implying that while local environment is driving flowering-time variation and flowering-time-related traits at large (Table 1), it only partly explains the early-/late-flowering partitioning of the phenotypes we observe among genetic lineages (Fig. 1c).

## Contribution of flowering-time genes to flowering-time-related trait variation

The genetic basis of flowering time has been extensively characterized at the molecular level in *B. distachyon* (e.g. Raissig and Woods 2022; Woods *et al.* 2023). In a second step, we also aimed to assess the contribution of genetic factors to flowering-time-related traits. Due to our relatively small sample size (56 accessions with phenotypes), we opted for a targeted approach rather than a classical GWAs and selected 22 flowering-time genes (Supplementary Table 5) molecularly characterized and described as impacting flowering time in our study system (Higgins *et al.* 2010; Wu *et al.* 2013; Woods *et al.* 2014, 2019, 2020; Sharma *et al.* 2017; Woods, Bednarek, *et al.* 2017; Woods, Ream, *et al.* 2017; Lomax *et al.* 2018; Qin *et al.* 2019; Cao *et al.* 2020; Kennedy and Geuten 2020).

Using the SNP calling performed by Minadakis *et al.* (2023), we extracted 502 SNPs across these 22 flowering-time genes and 56 accessions and performed stepwise RDA. We first fitted models where flowering-time-related traits were independently entered as response variable and each of the 502 SNPs were entered

**Table 1.** RDA output for each flowering-time-related trait and outdoor experiment.

| Trait | Model | Variable | Cumulative $R^2$ adj | AIC | F | P-value |
|---|---|---|---|---|---|---|
| MTD | Cluster alone | Cluster | 0.25 | 248.18 | 11.35 | 0.002 |
| | All bioclim variables | bio18 | 0.18 | 256.40 | 28.29 | 0.002 |
| | | bio16 | 0.22 | 250.64 | 7.81 | 0.006 |
| | | bio13 | 0.28 | 243.22 | 9.47 | 0.004 |
| | | All variables | 0.45 | | | |
| | Cluster + all bioclim variables | Cluster | 0.25 | 248.18 | 11.35 | 0.002 |
| | | bio18 | 0.42 | 218.35 | 34.53 | 0.002 |
| | | Elevation | 0.46 | 210.31 | 9.88 | 0.002 |
| | | All variables | 0.58 | | | |
| Days to flowerafter MTD | Cluster alone | Cluster | – | – | – | NS |
| | All bioclim variables | aridity_spring | 0.26 | 15.07 | 45.31 | 0.002 |
| | | bio9 | 0.30 | 10.32 | 6.77 | 0.006 |
| | | bio5 | 0.33 | 5.08 | 7.21 | 0.01 |
| | | bio18 | 0.37 | −0.46 | 7.46 | 0.006 |
| | | All variables | 0.52 | | | |
| Vernalization saturation | Cluster alone | Cluster | 0.10 | 126.14 | 4.31 | 0.004 |
| | All bioclim variables | aridity_spring | 0.22 | 105.17 | 35.61 | 0.002 |
| | | Elevation | 0.27 | 98.59 | 8.66 | 0.004 |
| | | All variables | 0.42 | | | |
| | Cluster + all bioclim variables | aridity_spring | 0.22 | 105.17 | 35.61 | 0.002 |
| | | Elevation | 0.27 | 98.59 | 8.66 | 0.006 |
| | | All variables | 0.44 | | | |
| Days to flower after vernalization saturation | Cluster alone | Cluster | 0.44 | 108.19 | 25.53 | 0.002 |
| | All bioclim variables | aridity_spring | 0.31 | 132.17 | 56.02 | 0.002 |
| | | bio16 | 0.39 | 116.82 | 18.17 | 0.002 |
| | | All variables | 0.54 | | | |
| | Cluster + all bioclim variables | Cluster | 0.44 | 108.19 | 25.53 | 0.002 |
| | | aridity_spring | 0.60 | 68.74 | 46.84 | 0.002 |
| | | bio13 | 0.65 | 51.19 | 19.98 | 0.002 |
| | | bio16 | 0.68 | 42.35 | 10.60 | 0.002 |
| | | srad_spring | 0.69 | 37.56 | 6.47 | 0.004 |
| | | All variables | 0.78 | | | |
| Day to flower (outdoor experiment) | Cluster alone | Cluster | 0.38 | 2,347.00 | 93.32 | 0.002 |
| | All bioclim variables | bio18 | 0.42 | 2,309.20 | 428.44 | 0.002 |
| | | bio12 | 0.45 | 2,276.00 | 36.18 | 0.002 |
| | | bio19 | 0.51 | 2,214.60 | 66.43 | 0.002 |
| | | All variables | 0.60 | | | |
| | Cluster + all bioclim variables | bio18 | 0.42 | 2,309.20 | 428.44 | 0.002 |
| | | Cluster | 0.48 | 2,243.40 | 19.46 | 0.002 |
| | | bio12 | 0.53 | 2,188.60 | 58.93 | 0.002 |
| | | bio19 | 0.56 | 2,146.50 | 45.21 | 0.002 |
| | | All variables | 0.66 | | | |

*Note.* bio18, precipitation of warmest quarter; bio9, mean temperature of driest quarter; bio5, max temperature of warmest month; bio13, precipitation of wettest month; bio16, precipitation of wettest quarter; bio12, annual precipitation; bio19, precipitation of coldest quarter. We only display predictors that increase the cumulative $R^2$ by more than 0.03 while "all variable" display the cumulative $R^2$ explained by all significant predictors together.

independently as explanatory variables. Here again, those models were compared to empty models where the respective flowering-time-related trait was explained by a fixed intercept alone. Three hundred and two SNPs across 20 flowering-time genes were significantly associated with at least one of the four flowering-time-related traits (Supplementary Table 6). Due to LD over short distances (Supplementary Fig. 4a), many SNPs gave precisely the same signal of association within a given gene (Supplementary Table 6). We thus pruned our dataset. Overall, the resulting 143 SNPs show a strong association with days to flower after vernalization saturation and MTD but very mild association with the two other traits (Fig. 2a). Individual SNPs explained as much as 42% of the variance in days to flower after vernalization saturation (Supplementary Table 6 and Fig. 2b). SNPs in *VERNALIZATION1* (*VRN1*), *FLOWERING LOCUS T-LIKE 1* and *10* (*FTL1* and *FTL10*), and *POLD3* show the strongest association with days to flower after vernalization saturation and MTD (Fig. 2, a and b). Interestingly, *VRN1* and *VERNALIZATION2* (*VRN2*), two genes known to play a role in vernalization requirement, show a stronger association with MTD than with vernalization saturation (Fig. 2a).

Because SNPs in flowering-time genes show a marginal association with vernalization saturation and days to flower after MTD, we focused on days to flower after vernalization saturation and MTD for the rest of the analysis. To further disentangle the ostensible confounding effect of population structure (Fig. 2b), SNPs in flowering-time genes, and environmental variables on days to flower after vernalization saturation and MTD, we fitted additional models that included the phenotype as a response variable and (i) the cluster of origin alone (model = cluster alone), (ii) all the 20 SNPs (1 SNP per gene; Supplementary Fig. 4b) showing the strongest association with either days to flower after vernalization saturation or MTD (model = all SNPs), and (iii) the cluster of origin, all the 20 SNPs, and the environmental variables associated with each respective trait (as shown in Table 1) as explanatory variables (model = cluster + all SNPs + envt.). We compared these models successively with the empty model as previously described.

For both traits, single SNPs (e.g. Bd1_5866489 in *VRN1* for days to flower after vernalization saturation and Bd2_5415945 in FTL1 for MTD) gave separately identical or similar association as the cluster of origin alone (Table 2). This results in a reduced or null cluster effect in the full model compared to the one of single

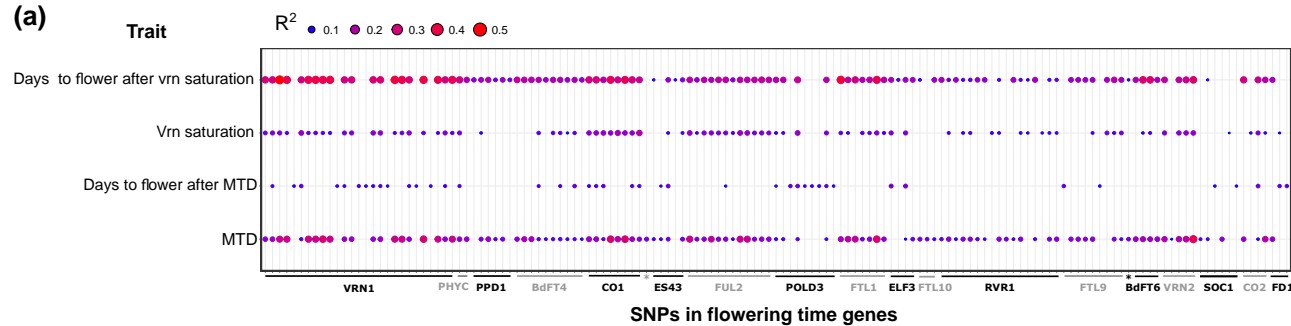

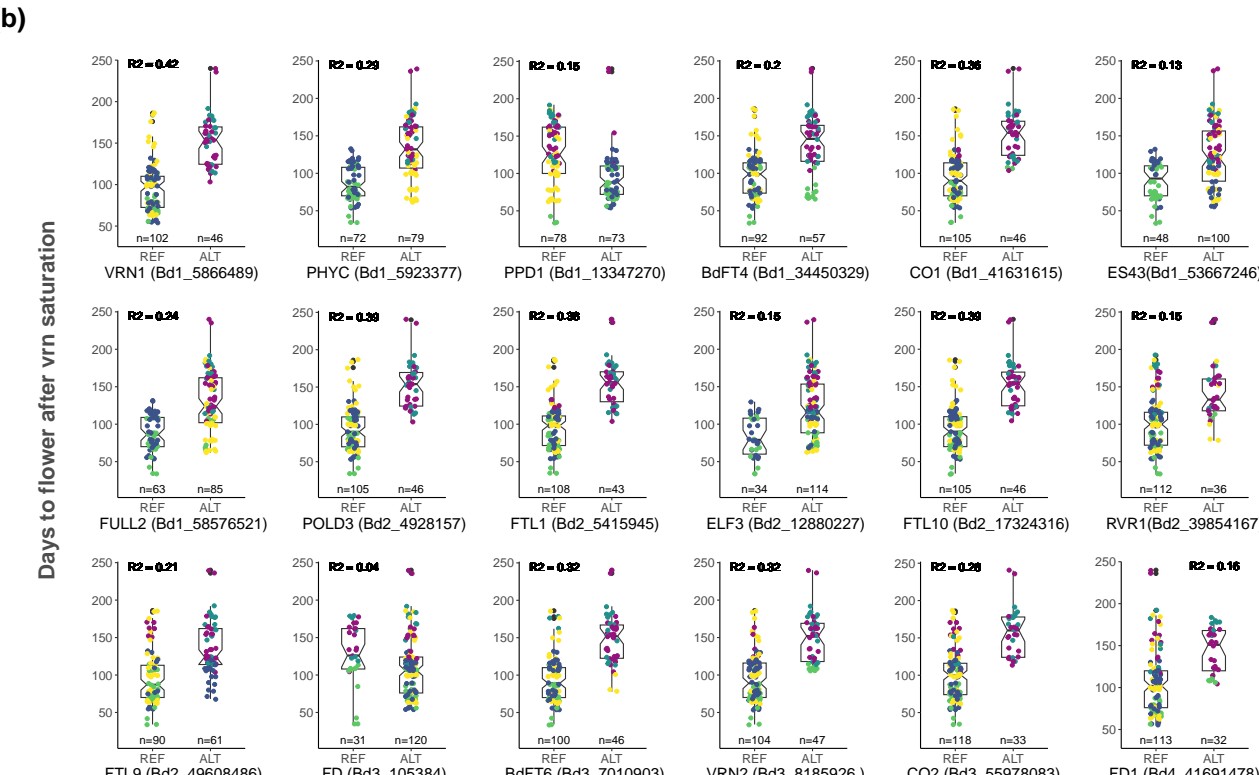

**Fig. 2.** Association between SNPs in flowering-time genes and flowering-time-related traits. a) RDA output for the 143 pruned SNPs. b) For each gene, the boxplots contrast the effect of the reference (Bd21) and alternative alleles on days to flower after vernalization saturation for the SNP showing the strongest association with the trait. The color code is the same as in Fig. 1.

SNPs (Table 2). Note that the variance explained by a given SNP (e.g. Bd1_5866489 in *VRN1*) differs from the one obtained above (Fig. 2b). RDA do not allow missing values and the models with 20 SNPs (Table 2) are therefore based on a slightly different sample size than the model performed with single SNP (Fig. 2b) which results in different $R^2$ estimates.

As mentioned above, RDA optimize the variance explained but do not allow to identify the mechanistic driver of variation. Considering the allele distribution of SNPs in flowering-time genes across genetic cluster (Fig. 2b), disentangling and quantifying the effect of population structure from the one of genetic factors or single genes remains therefore challenging for most SNPs in flowering-time genes.

### Differentiation of flowering-time genes

Most SNPs in flowering-time genes nonetheless appear to be highly differentiated among genetic clades. This could result from two processes. On the one hand, *B. distachyon* populations underwent bottlenecks during the last glaciation (Minadakis *et al.* 2023), which may have led to reduced genetic diversity and highly differentiated alleles among genetic lineages/clades genome-wide (neutral scenario). On the other hand, flowering-time genes might be under positive selection and therefore highly differentiated among genetic clades (selection scenario).

To first assess the extent to which our SNPs of interest are more differentiated compared to genome-wide levels, we computed single SNP $F_{ST}$. To limit the number of comparisons and considering the partitioning of the reference and alternative alleles among accessions in the flowering-time genes (Fig. 2b), we only computed $F_{ST}$ between accessions of the A and B lineages. We found that the large majority of SNPs significantly associated with flowering time are above the 3rd quartile of the genome-wide distribution or even belong to the top 5% outliers (Fig. 3a). Hence, with the exception of SNPs in *EARLY FLOWERING 3* (*ELF3*), *FD*, and *FD1*, all flowering-time genes harbor SNPs that tend to be more differentiated than the rest of the genome.

**Table 2.** Output of the RDA between flowering-time-related traits, SNPs in flowering-time genes, and bioclimatic variables.

| Trait | Model | Variable | Cumulative $R^2$ adj | AIC | F | P-value |
|---|---|---|---|---|---|---|
| Days to flower after vernalization saturation | cluster alone | Cluster | 0.56 | 10.72 | 26.44 | 0.002 |
| | All SNPs | Bd1_5866489 | 0.56 | 7.58 | 103.39 | 0.002 |
| | | Bd3_8185926 | 0.58 | 4.51 | 5.04 | 0.01 |
| | | All variables | 0.68 | | | |
| | Cluster + SNPs + aridity_spring + bio13 | Bd1_5866489 | 0.56 | 7.58 | 103.39 | 0.002 |
| | | aridity_spring | 0.65 | −10.08 | 21.41 | 0.002 |
| | | bio13 | 0.69 | −18.38 | 10.44 | 0.002 |
| | | Bd2_5415945 | 0.72 | −25.66 | 9.22 | 0.002 |
| | | Bd4_41691478 | 0.75 | −35.32 | 11.62 | 0.004 |
| | | | 0.87 | | | |
| MTD | cluster alone | Cluster | 0.32 | 132.53 | 10.57 | 0.002 |
| | All SNPs | Bd2_5415945 | 0.37 | 123.86 | 48.04 | 0.002 |
| | | All variables | 0.57 | | | |
| | Cluster + SNPs + elevation + bio18 | Bd2_5415945 | 0.37 | 123.86 | 48.04 | 0.002 |
| | | bio18 | 0.46 | 112.20 | 14.32 | 0.002 |
| | | Cluster | 0.52 | 105.56 | 3.66 | 0.008 |
| | | Bd3_8185926 | 0.57 | 98.44 | 8.71 | 0.006 |
| | | All variables | 0.61 | | | |

To gain power for our analyses, we further made use of the *B. distachyon* diversity panel composed of 332 natural accessions (Minadakis *et al.* 2023; Fig. 4a) and found very similar levels of allele differentiation at flowering-time genes (Supplementary Fig. 4c). These results are in line with a forward simulation we performed under a neutral scenario using the demographic estimates computed by Minadakis *et al.* (2023). The $F_{ST}$ distribution calculated between the simulated A and B lineages, as the distribution obtained with the real data, is indeed largely shifted toward low values (Fig. 3b). These latter results demonstrate that bottlenecks did not lead to elevated genetic differentiation at the genome-wide level. As such, the high $F_{ST}$ value we observed suggests that most flowering-time genes are under selection.

## Genome-wide scans of positive selection

To properly test for positive selection while accounting for the structure and demographic history of our populations, we computed $X^TX$ statistics, a measure comparable to single SNP $F_{ST}$ that accounts for the neutral covariance structure across populations. In brief, we computed $X^TX$ with our actual SNP dataset over the entire diversity panel using the five genetic clades as focal populations. We then simulated a POD of 100,000 SNPs under the demographic model inferred from the covariance matrix of the actual SNP dataset. $X^TX$ statistics were then computed for the POD to determine the probability of neutrality for each SNP. The threshold of significance was thus set to 11.2, a value slightly lower than the 1% outlier threshold (13.6). Many flowering-time genes display SNPs more differentiated than expected under a neutral scenario (Fig. 3b) suggesting that those genes have evolved under positive selection. Nine flowering-time genes PHYTOCHROME C (PHYC), VRN1, CONSTANS 1 and 2 (CO1 and CO2), POLD3, FTL1, FTL10, FTL9, and VRN2 display extremely differentiated SNPs ($X^TX$ > 15) both in the subset of accessions (Fig. 3a) and in the entire diversity panel (Supplementary Fig. 4c). We also tested for extended haplotypes and footprint of selective sweeps with the iHS. However, none of those highly flowering-time genes are located in regions displaying significantly longer haplotypes (Supplementary Fig. 5).

We ran SnpEff to test which SNPs in flowering-time genes are more likely to have a functional impact. Only few SNPs in CO2, ES43, and Photoperiod-H1 (PPD1) were categorized as variants with high impact while the large majority of SNPs in other genes were categorized as variants with moderate and low effect (Supplementary Fig. 6), rendering the identification of the potential targets of selection challenging for most of our genes of interest.

In order to characterize which environmental factors might have shaped diversity at flowering-time genes, we eventually made use of genotype-environment association (GEA) analyses performed by Minadakis *et al.* (2023) with bioclimatic variables associated to precipitation levels, temperature, or elevation. We found that only one of the 22 flowering-time genes (Bradi2g59119, *ODDSOC2-like*) showed an overlap with the gene sets significantly associated with current bioclimatic variables (Fig. 3c).

Yet, the GEAs performed by Minadakis *et al.* (2023) corrected for population structure and can hence result in false negative since the genetic clades occupy different ecological niches. We indeed found significant associations between SNPs at flowering-time genes and bioclimatic variables such as Bio14 (precipitation of driest quarter) and aridity levels in spring, when not accounting for population structure (Kendall correlation; Supplementary Fig. 7a), indicating that the confounding effect of population structure and adaptation at a regional scale may mask the significant effect of the environment on flowering-time genes diversity.

## Potential adaptation to past climatic conditions

We estimated the age of flowering-time alleles and found that those arose relatively recently between 9,000 and 38,000 years ago (Supplementary Table 7) as most alleles in our system (Minadakis *et al.* 2023). We speculated that variation in flowering time could also reflect adaptation to recent past conditions, potentially to the LGM 22,000 years ago. We therefore tested whether the delay in flowering time we observed in accessions from the A_East and A_Italia clades could result from an adaptation to past colder climates. To do so, we used the niche suitability projections under LGM conditions computed by Minadakis *et al.* (2023). We selected a set of 200 random points per clade in highly suitable habitats and extracted the 19 LGM bioclimatic variables for the corresponding sites. The PCA performed with these 19 LGM bioclimatic variables does not allow us to separate the five genetic clades. Altogether, accessions from the A_East and A_Italia clades neither occurred in colder nor in wetter environments than the B_East, B_West, and C accessions (Supplementary Fig. 8) which suggests that the extended vernalization requirement and delay in

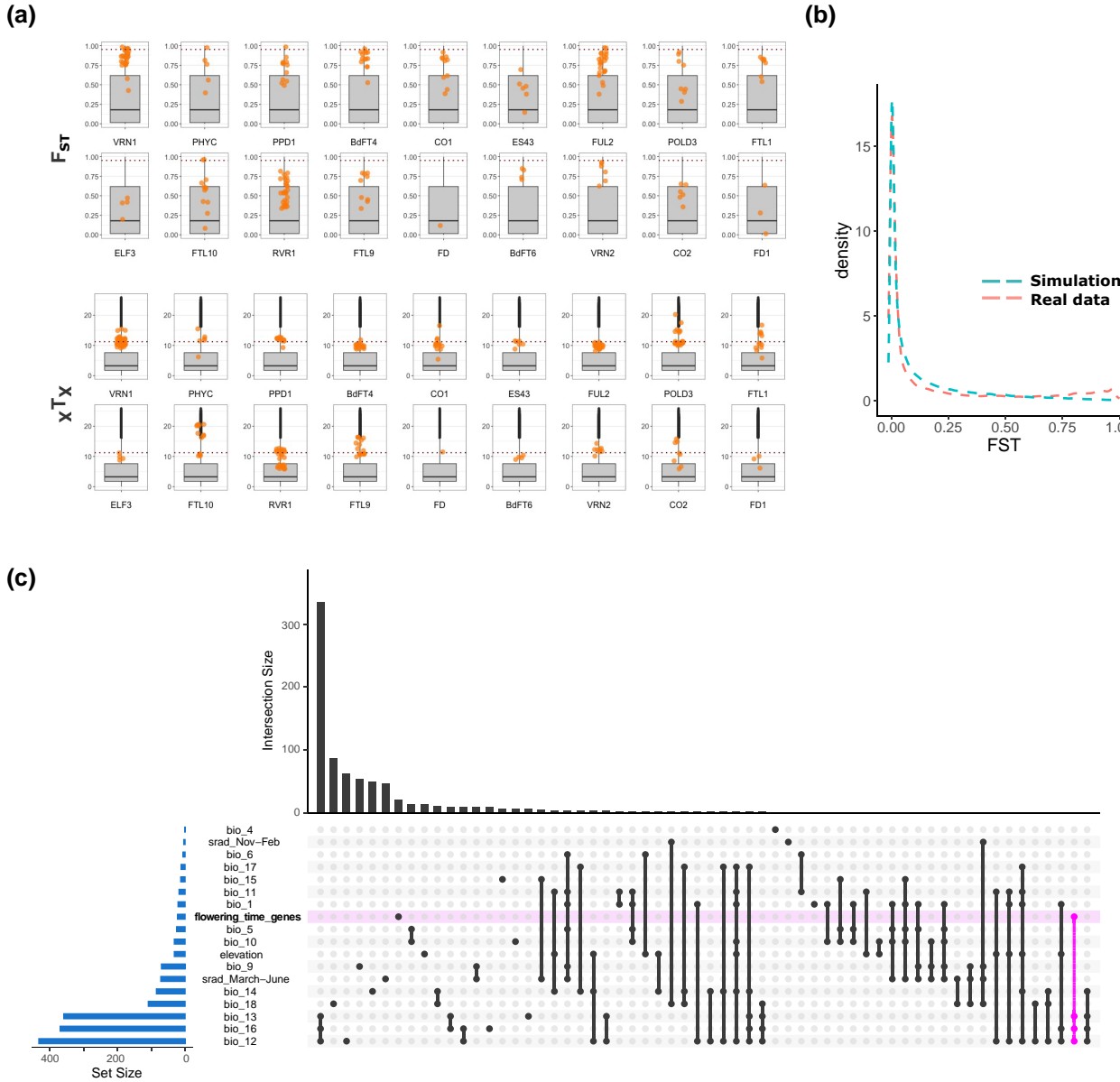

**Fig. 3.** Flowering-time gene differentiation: a) Top: single SNP $F_{ST}$ computed for the 56 accessions of a and b lineages (dashed line = 5% outliers); bottom: single SNP $X^T X$ computed at the genetic clade level (dashed line = 1% POD threshold). b) Distribution of $F_{ST}$ calculated between the A and B lineage with real and forward-simulated data under a neutral scenario. c) Upset plot displaying the overlap between gene sets identified with the GEAs performed by Minadakis *et al.* (2023) for 23 bioclimatic variables and flowering-time genes.

flowering we observed for the A_East and A_Italia accessions were not selected during LGM.

## Long-range LD among flowering-time genes

A striking pattern eventually emerges from the former analyses. The fact that the flowering-time genes gave nearly identical signals with regard to their association with flowering time suggests, as displayed in Fig. 2b, that SNPs at flowering-time genes harbor similar allele frequencies. These similar allele frequencies indicate that some flowering-time genes might be co-selected (polygenic selection) and remain in strong LD despite the large physical distances that separate them (Zan and Carlborg 2019; Gupta *et al.* 2023).

To formerly test this hypothesis, we computed LD among the 20 focal SNPs used for the RDA analyses. LD computed among pairs of focal SNPs in the 56 selected accessions as well as in the

entire diversity panel show a similar pattern: many flowering-time genes display high LD despite the large distances that separate them on a given chromosome (Fig. 4b). This is especially true for flowering-time genes located on chromosomes 1 and 2 (Bd1 and Bd2), which are found in much higher LD than expected considering the 95% CI of the genome-wide LD decay (Fig. 4c).

Strikingly, we also found very high LD among flowering-time genes located on different chromosomes both in the selected accessions and entire diversity panel (Fig. 4b). Located on chromosome 1, VRN1, for instance, is in strong LD with eight other genes located on chromosome 2 (POLD3, LD = 0.84; FTL9, LD = 0.45; FTL1, LD = 0.76; FTL10, LD = 0.83), chromosome 3 (VRN2, LD = 0.65; CO2, LD = 0.55, BdFT6, LD = 0.54), and chromosome 4 (BdFT4, LD = 0.71). To test to what extent this long-range LD among the 14 focal flowering-time genes deviates from genome-wide patterns, we computed LD for a subset of 50,000 random

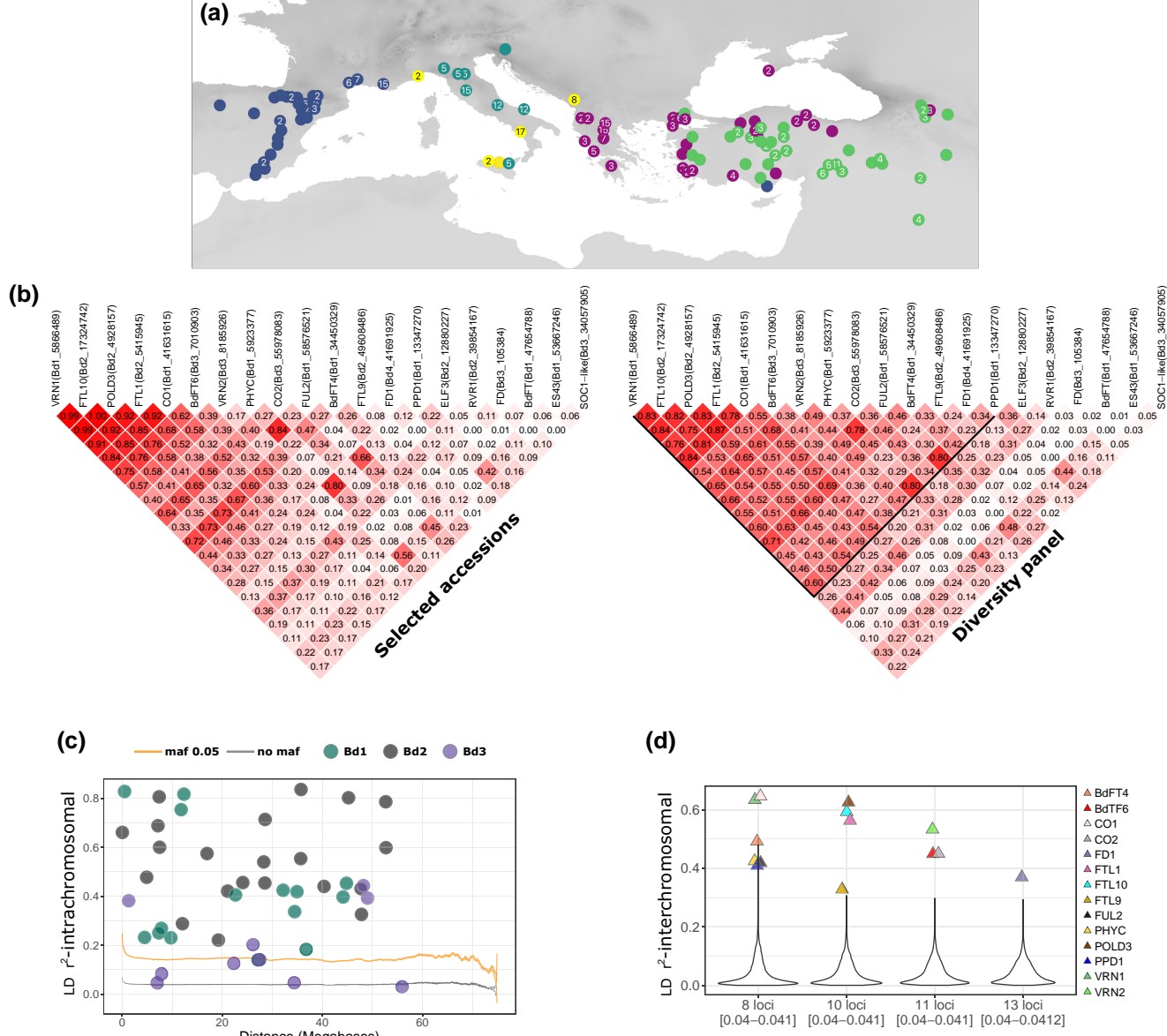

**Fig. 4.** LD in the flowering-time pathway. a) Map displaying the locality of the 332 sequenced accessions composing the diversity panel. b) LD among flowering-time genes in the selected accessions and diversity panel. For the diversity panel, the black lines delimit the focal genes used for the inter-chromosomal LD analysis. c) LD decay computed over the entire diversity panel. The orange and gray lines display the 95% CI of the LD decay calculated with or without filtering for a minimum allele frequency (maf) respectively. Dots display LD between pairs of flowering-time genes located on the same chromosome. d) Distribution of LD among loci located on different chromosomes. The violin plots display the means of LD calculated among eight, 10, 11, and 13 loci. Numbers in square brackets indicate the CI around the mean.

pairs of genic loci located on different chromosomes across the genome. This allowed us to estimate the average inter-chromosomal LD among loci (Fig. 4d). For each combination, the inter-chromosomal LD we observed among eight loci (for focal flowering-time genes located on chromosome 1), 10 loci (for focal flowering-time genes located on chromosome 2), 11 loci (for focal flowering-time genes located on chromosome), and 13 loci (for focal flowering-time genes located on chromosome 4) was largely outside the CI calculated with random genic loci (Fig. 4d).

Note that *FRUITFULL-like* (*FUL2*) and *PPD1* are the only genes displaying putative footprint of insertion polymorphisms in the diversity panel (Supplementary Table 5). Furthermore, in a selfing species like *B. distachyon*, heterozygous SNPs are hallmarks of gene duplication (Stritt *et al.* 2022; Jaegle *et al.* 2023). Using a vcf file not filtered for heterozygous SNPs, we found that only 0.6%

of the SNPs located in flowering-time genes are heterozygous across the 332 accessions. These results imply that flowering-time genes are occurring in single copy and that structural rearrangements or duplications did not alter their position in non-reference genomes. Taken together, these results indicate that the long-range LD among flowering-time genes as well as their differentiation levels deviate from genome-wide patterns and are difficult to explain by the demographic history of the population alone. We hence conclude that 14 flowering-time genes (Fig. 4d) are co-evolving and undergo polygenic selection.

## GWA for flowering-time variation measured in outdoor conditions

Greenhouse conditions are far from any natural optimum and may obscure association with relevant environmental cues, while

taking an environment closer to the natural conditions encountered by one or several clades acts as a useful pivot to contrast clades. Hence, we eventually made use of flowering-time data we collected for a subset of 131 accessions (Stritt *et al.* 2022) in Zürich, Switzerland, to assess flowering-time variation in semi-natural conditions. While *B. distachyon* does not occur at such latitudes (Minadakis *et al.* 2023), a PCA performed with the 22 bioclimatic variables on our 332 natural accessions and including a site in Zürich shows that the environmental conditions in Zürich are not different from the ones encountered by ABR9 for instance (Fig. 5a and Supplementary Table 2 for bioclimatic data). While other biotic (vegetation type and density) and abiotic (soil characteristic) factors are certainly preventing the species from occurring in northern latitude, Zürich harbors climatic conditions similar to ones encountered by the species at the northern margin of its natural distribution, making it a valid experimental site to study flowering time in *B. distachyon*.

In brief, we planted seeds for 131 accessions (Fig. 5b) outdoors in November 2017, in the Botanical Garden of Zürich, Switzerland, with six replicates per accession and recorded flowering time in spring. Thirty-one of the accessions used in the greenhouse experiment are used in the outdoor experiment, including four (Ren4, Lb1, Lb23, and Cm7) of the five accessions that did not flower by the end of the experiment. All plants flowered within 20 days in April (Fig. 5c). Flowering time for plants from the B_East and C clades were not significantly different (Kruskal–Wallis test, *P*-value = 0.96) but those accessions flowered significantly faster than accessions from the other clades (Kruskal–Wallis test, all *P*-values < 0.01). Plants from the A_Italia and B_West flowered significantly later than plants from the other clades (Kruskal–Wallis test, all *P*-values < 0.02) but were not significantly different from each other (*P*-value = 0.06). As such, the data collected outdoor contrast with the ones collected in the greenhouse.

As for the greenhouse experiment, we used RDA to fit models where flowering time was independently entered as a response variable while the cluster of origin (model = cluster alone), the 22 environmental variables (model = all bioclimatic variables), or both the cluster of origin and the 22 bioclimatic variables (model = cluster + all bioclimatic variables) were successively entered as explanatory variables. We compared those full models to empty models where flowering time was explained by a fixed intercept alone (Capblancq and Forester 2023). We found that a large part of the variance in flowering time is explained by bioclimatic variables linked to precipitation levels in warm months whether the models did or did not include the cluster of origin (e.g. bio18 and bio12; Table 1).

A GWAs performed with GEMMA (Zhou and Stephens 2012) on the 131 accessions and 2,266,225 filtered SNPs (Supplementary Fig. 9 for marker density) identified one significant peak which does not overlap with any of the 22 flowering-time genes we studied above (Fig. 5d). We also made use of the GEAs performed by Minadakis *et al.* (2023) and found that the gene underlying the peak (Bradi2g11490, a carbohydrate-binding-like gene) does not overlap with the gene sets significantly associated with current bioclimatic variables. Yet, we found significant associations (Supplementary Fig. 7b) between SNPs at the GWAs peak and bioclimatic variables, especially with Bio14 (Precipitation_of_Driest_Month), when not taking into account population structure. Consistent with selection by the environment, the top GWAs SNP constitutes also an $X^TX$ outlier (Fig. 5e).

## Discussion

### Flowering-time-related traits and adaptation to local climate

Flowering time has been shown to play an important role in local adaptation across many plant systems including crops (Izawa 2007; Anderson *et al.* 2012; Anderson, *et al.* 2013; Ågren *et al.* 2017; Navarro *et al.* 2017; Wadgymar *et al.* 2017; Takou *et al.* 2019; Qian *et al.* 2020; Yan *et al.* 2021). In *B. distachyon*, flowering time has been mainly studied from a molecular viewpoint, clarifying the role of specific genes in sensing environmental variation such as day-length and temperature (Ream *et al.* 2012, 2014; Ruelens *et al.* 2013; Woods *et al.* 2014, 2020; Woods, Bednarek, *et al.* 2017; Woods, Ream, *et al.* 2017; Lomax *et al.* 2018; Cao *et al.* 2020; Kennedy and Geuten 2020; Bouché *et al.* 2022; Raissig and Woods 2022). While those studies underline the complex architecture of flowering time and revealed large variation regarding vernalization requirement among natural accessions for instance, they provide little information with regard to adaptation.

In fact, the wide range of vernalization requirement observed in *B. distachyon* suggests that local adaptation of flowering time could take place through distinct developmental thresholds (i.e. the accumulated amount of "developmental time" required for a given developmental stage to allow a transition to the next developmental stage, see Donohue *et al.* 2014) among accessions. Indeed, in a widespread Mediterranean species like *B. distachyon*, not all accessions are exposed to long and harsh winter (Supplementary Fig. 2). The expression of traits such as the minimum duration of vernalization necessary to permit flowering might be contributing *in fine* to flowering-time variation in the wild and be especially relevant to investigate in the context of local adaptation. We indeed observed significant albeit partial correlations among the four flowering-time-related traits we investigated (Fig. 1b) indicating that these traits constitute different components of flowering time.

Estimating the effect of the environment remains a challenge in such structured populations. Yet, we show a significant association between flowering-time-related traits and bioclimatic variables in addition to the cluster effect, indicating that flowering-time-related traits are locally adapted in our system. Because bioclimatic variables are largely correlated among each other, pinpointing the selective constraints acting on flowering-time-related traits is equally difficult (Capblancq and Forester 2023). Our RDA analyses, however, point at bioclimatic variables which are biologically meaningful as linked to aridity levels in warm months. As such, precipitation levels and hence water limitation seem especially important for flowering-time-related traits as already shown for broader fitness effects in our species (Des Marais *et al.* 2017). In our system, MTD and days to flower after MTD have never been investigated. It is therefore very interesting to see that those two traits also display signs of local adaptation. Our results therefore advocate for a complex adaptation of flowering time and suggest that vernalization saturation might not be necessary in the wild to synchronize flowering time with the environment.

### Early- and late-flowering genotypes are unlikely driven by adaptation

The greenhouse experiment further confirms that *B. distachyon* natural accessions vary greatly regarding their vernalization

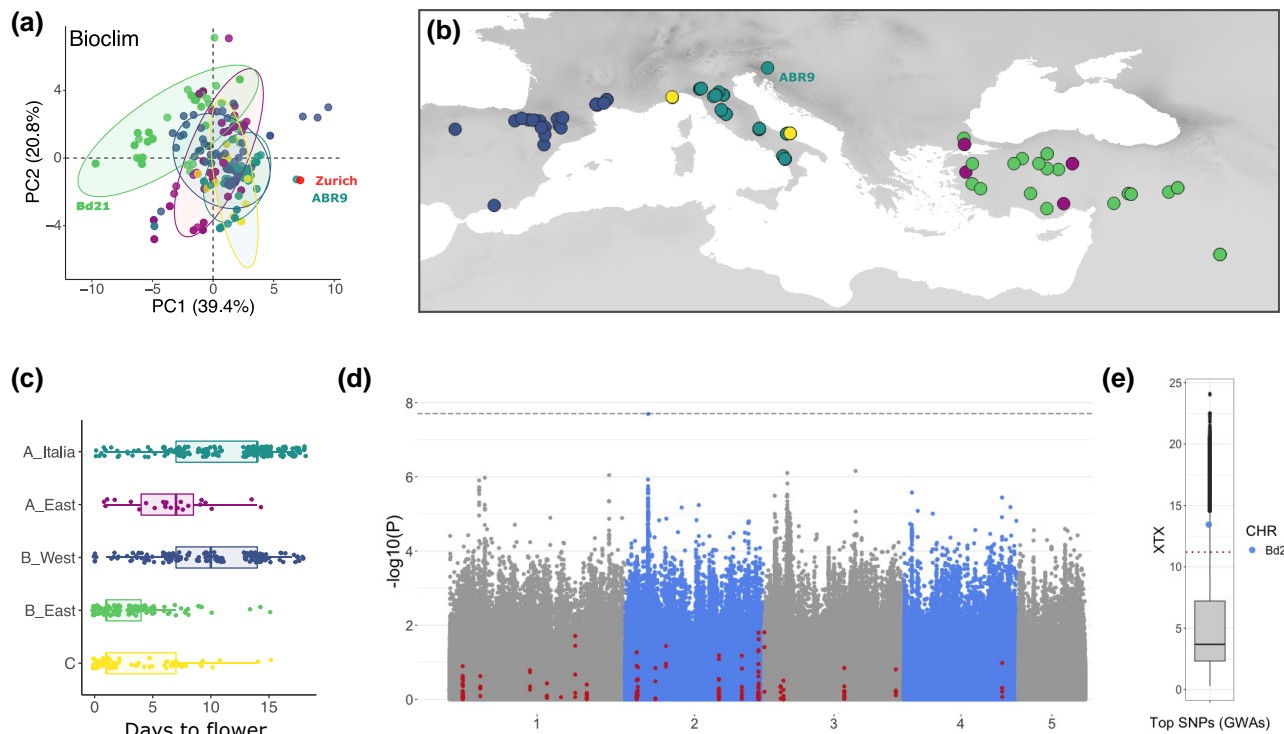

**Fig. 5.** GWA with flowering time (outdoor experiment). a) PCA performed with 19 classical worldclim variables combined to solar radiation in spring, global aridity in spring, and altitude for the 332 accessions of the diversity panel and a locality in Zürich. b) Geographical origin of the 131 accessions used for the experiment. c) Flowering time per cluster (displaying replicates). d) Manhattan plot displaying the association between SNPs and flowering time. The red dots display SNPs located into flowering-time genes. The dashed line corresponds to FDR threshold of significance. e) $X^TX$ values for the top GWAs SNP on chromosome 2 (Bd2).

saturation requirement and ultimately flowering time and can be broadly grouped into early- and late-flowering genotypes (Supplementary Fig. 1 and Fig. 1, but see Ream *et al.* 2014; Gordon *et al.* 2017 for a finer classification scheme). Despite within clade variation regarding flowering time after vernalization saturation (Fig. 1c), accessions from the A lineage (A_East and A_Italia) display a large delay in flowering compared to accessions from the B (B_East and B_West) and C lineage, which typically results from longer vernalization saturation requirements. This pattern has been suggested to be a sign of adaptation at a regional scale in Turkey (Skalska *et al.* 2020) and could be interpreted as a footprint of diversifying selection at the genetic clade level. Based on this hypothesis, accessions from the A_East and A_Italia clades may display a delay in flowering as globally adapted to colder and less arid environmental conditions, as shown for instance in *A. thaliana* (Ågren and Schemske 2012). Yet, our results do not fully support this scenario, as we only find partial correlations between the partitioning of the phenotypes and the bioclimatic variables associated with flowering time (Fig. 1 and Supplementary Fig. 3). Our niche modeling projections for the LGM (Minadakis *et al.* 2023) do not further support an adaptation to the LGM conditions (Supplementary Fig. 8). As such, the partitioning of the early- vs late-flowering accessions remains difficult to explain with an "adaptive" scenario. In summary, we conclude that local environmental conditions are partly driving flowering-time variation within genetic clades but not the early- and late-flowering partitioning of the phenotypes we observed among clades in the greenhouse.

We have not tested different combinations of vernalization temperature and length. In *A. thaliana* for instance, natural ecotypes have characteristic vernalization temperature profiles and

vernalization temperature ranges from 0 to 14 °C (Duncan *et al.* 2015). While *B. distachyon* does not occur in a wide range of latitudes as *A. thaliana*, it is clear that interactions between vernalization temperature and length are likely to play a role as well. As already shown by Stritt *et al.* (2022) with a smaller subset of accessions, variation in flowering time is for instance largely attenuated in outdoor conditions and although significant differences are observed among genetic clades, all plants flowered within 20 days when grown outdoors in Zürich (Fig. 5c), including the four accessions that did not flower with the greenhouse experiment. Hence, greenhouse experiments, as largely performed in *B. distachyon* (Ream *et al.* 2014; Gordon, *et al.* 2017; Sharma *et al.* 2017), may only weakly capture how accessions behave in the wild.

## Flowering-time genes and flowering-time variation

We also selected 22 flowering-time genes known to play a major role in vernalization- (e.g. *VRN1* and *POLD3*) or photoperiod sensing (e.g. *PPD1* and *PHYC*) to characterize the magnitude of their allelic effects on flowering-time-related traits. We found that SNPs in those 22 genes display relatively large association with the number of days to flower after vernalization saturation and to some extent with MTD or vernalization saturation but only poorly with the number of days to flower after MTD (Fig. 2a). Interestingly, genes known to play a role in cold-mediated vernalization such as *VRN1* or *POLD3* (Raissig and Woods 2022) show a larger association with MTD than with vernalization saturation, suggesting that those genes are important for cold perception regardless of its length.

Here again, our RDA analyses show that disentangling the effect of the cluster of origin from the one of single SNPs is

challenging. We indeed observed a strong partitioning of the reference and alternative alleles in flowering-time genes among genetic clades and lineages (Fig. 2b). The high genetic differentiation of flowering-time genes between the A and B lineages (Fig. 2b) could result from the recent bottlenecks experienced by *B. distachyon* in the recent past (Stritt *et al.* 2018; Minadakis *et al.* 2023) and produce spurious association with flowering-time-related traits. Three lines of evidence rule out this demographic scenario.

First, our 22 candidate genes have been shown to play an important role in photoperiod sensing, vernalization, and *in fine* in flowering variation among diverse *B. distachyon* accessions not only by sequence homology but also via molecular, biochemical, and genetic methods (Wu *et al.* 2013; Ream *et al.* 2014; Woods *et al.* 2014, 2019, 2020, 2023; Woods, Bednarek, *et al.* 2017; Woods, Ream, *et al.* 2017; Lomax *et al.* 2018; Qin *et al.* 2019; Bouché *et al.* 2022; Raissig and Woods 2022; Alvarez *et al.* 2023). It is therefore legitimate to assume that they might play a role in regulating flowering-time-related traits in genetically diverse accessions as well.

Second, $F_{ST}$ analyses performed with our real- as well as forward-simulated data show that the population size reduction experienced by *B. distachyon* did not lead to genome-wide highly differentiated alleles and SNPs at flowering-time genes constitute clear $F_{ST}$ outliers. The $X^{T}X$ analysis we performed with the five genetic clades as focal populations also shows that most of the flowering-time genes harbor SNPs above the threshold of significance, *POLD3* and *CO1* presenting more extreme outliers. This footprint of positive selection (Gautier 2015) is yet not accompanied by extended haplotypes around our candidate genes (except for *CO2*) which implies that the initial selective sweeps may have eroded with time and that selection did not occur in a recent past. Flowering-time genes are yet not colocalizing with regions we previously identified with GEA analyses (Fig. 3c). Thus, although we find evidence of positive selection at single flowering-time genes, the selective constraint at play remains yet to be identified. It is yet essential to keep in mind that GEA performed in species with strong population structure can lead to high rate of false negatives as they typically include structure correction (Booker *et al.* 2023; Lottheros 2023). We did find significant associations between bioclimatic variables and SNPs at flowering-time genes when not correcting for population structure (Supplementary Fig. 7). Common garden experiments will thus be key to test the effect of genotype-by-environment interactions in our system.

Third, we highlighted a striking pattern of long-range LD, both intra- and inter-chromosomal, among 14 flowering-time genes associated with flowering-time variation under greenhouse conditions (Fig. 4). Polygenic selection is expected to result in LD between regions under selection (Yeaman *et al.* 2016, 2018; Gupta *et al.* 2023) but only few cases have been reported so far in plants and animals (Hohenlohe *et al.* 2012; Yeaman *et al.* 2016; Park 2019; Gupta *et al.* 2023). In *A. thaliana*, Zan and Carlborg (2019) also identified long-range LD among four clusters of flowering-time genes. We demonstrated that the levels of long-range LD (intra- and inter-chromosomal) we observed among 14 flowering-time genes (Fig. 4) are not observed among random genic loci in the genome. As flowering-time genes do not display signs of duplication or insertion polymorphisms, we imply that structural rearrangements did not bring flowering-time genes physically together in non-reference accessions. We therefore believe that we present clear evidence for polygenic selection on key genes involved in the flowering pathway. Expression analyses further support the functional connection among these loci in *B. distachyon* and grasses in general. For example, *VRN1* and *FTL1* are expressed in a positive feedback loop, which overcomes the flowering repression of *VRN2* (Ream *et al.* 2014; Woods *et al.* 2016; Woods, Ream, *et al.* 2017). Additionally, there is an intricate connection between many of these flowering-time genes both transcriptionally and at the protein level. For example, among the pairwise interactions tested between *PHYC*, *PHYB*, *ELF3*, *PPD1*, *VRN2*, *CO1*, and *CO2* more than 80% showed positive interactions in yeast two hybrid assays some of which have been verified *in planta* (Shaw *et al.* 2020; Alvarez *et al.* 2023). Thus, many flowering-time genes in LD can interact at multiple levels.

Altogether, this set of analyses show that the association between flowering-time genes and flowering-time-related traits, especially days to flower after vernalization saturation, is a result of selection and unlikely due to population structure. Interestingly, while *POLD3* and *ELF3* loss of function mutants flower faster than the wildtype (Woods *et al.* 2020; Bouché *et al.* 2022), their contribution to flowering time after vernalization saturation for instance (39 and 15% respectively) differ drastically. This discrepancy might be due to the fact that mutants used to characterize gene functions usually harbor mutations with deleterious or large effect size (loss of function). In contrast, our natural flowering-time gene variants are mostly predicted as having low to moderate suggesting that they modulate flowering-time-related traits quantitatively. As such, one should remain cautious while extrapolating the effect of flowering-time variants from mutant-based experiments.

## GWAs detect additional candidate genes for flowering time

None of the flowering-time genes colocalize with the GWAs candidate identified with the outdoor experiment (Fig. 5d). It is clear that the partitioning of the reference and alternative variants at flowering-time genes (Fig. 2b) largely diminishes our detection power as we applied a correction for population structure. Yet, while we should extend our common garden experiment to a larger number of sites, our results are in line with a previous study in *A. thaliana* which, by using natural accessions and RILs, found that flowering-time variation scored in the field experiment poorly correlated with the flowering-time variation obtained under greenhouse conditions (Weinig *et al.* 2002; Malmberg *et al.* 2005; Brachi *et al.* 2010; Wilczek *et al.* 2010). As a consequence, a limited overlap is observed between the genomic regions detected in field experiments and those detected under greenhouse conditions (Brachi *et al.* 2010). Flowering time has a complex polygenic architecture (Buckler *et al.* 2009; Brachi *et al.* 2010; Navarro *et al.* 2017; Zan and Carlborg 2019; Gaudinier and Blackman 2020) and the use of EMS-induced mutants suggested that many additional genes might play a role in shaping this trait in *B. distachyon* (Raissig and Woods 2022), some of which are not described or only play a minor role in flowering time in other plant models (Woods *et al.* 2020). In addition, alleles may affect phenotypes only in specific populations (Zan and Carlborg 2019; Yan *et al.* 2021; Gloss *et al.* 2022) or seasons (Weinig *et al.* 2002; Gould and Stinchcombe 2017). Taking into account the polygenic architecture, gene-by-environment association and phenotypic plasticity (Gaudinier and Blackman 2020; Yan *et al.* 2021) will therefore be essential to better capture the adaptive potential of flowering time and flowering-time genes in natural populations of *B. distachyon*. Although flowering-time genes are undoubtedly essential in the perception of environmental cues and overwintering (for review Raissig and Woods 2022), our results show that the effect of their variants in the wild are more difficult to predict and that additional genes may be at play.

## Conclusion and perspectives

Our results suggest that (i) flowering-time-related traits are locally adapted in *B. distachyon* but (ii) part of the variation in those traits can be explained by the environment and by SNPs in key flowering-time genes, and (iii) key flowering-time genes are co-evolving but their effect in the wild remains to be clarified. In the face of global warming, plant phenology has recently advanced significantly (e.g. Anderson *et al.* 2012). Investigating the polygenic architecture of flowering time therefore remains a timely question. Polygenic selection, epistatic and pleiotropic effects might limit the evolution of traits (Yan *et al.* 2021; Yeaman 2022) and future experiments in *B. distachyon* should focus on disentangling these effects. Mimicking the combination of the various clues that trigger flowering is yet impractical under greenhouse conditions and common garden experiments will thus be essential to place flowering time in a natural context in this system. We, however, know relatively little about the basic ecology of *B. distachyon*. Fundamental questions now need to be addressed to more broadly characterize the process of local adaptation in this species. These include: when are plants emerging in the wild?; how plastic are flowering time and phenotypes in general?; and what is the contribution of seed banks to the effective population size and hence selection strength? Answering these will undoubtedly represent remarkable progress in our understanding of *B. distachyon's* ecology.

## Data availability

Seeds will be distributed through GRIN (https://www.ars-grin.gov) or can be provided in small quantity upon request. Accession numbers for sequencing data can be found in Gordon *et al.* (2017, 2020), Skalska *et al.* (2020), Stritt *et al.* (2022), and Minadakis *et al.* (2023). The raw data of the flowering-time measurements are available in the Supplementary material. Bioclimatic variables can be downloaded at https://www.worldclim.org/data/bioclim.html and extracted using the R code "download_worldclim2.1.R" provided by Minadakis *et al.* (2023) (https://zenodo.org/records/8354318). RDA analyses were adapted from the detailed scripts provided by Capblancq and Forester (2023; https://github.com/Capblancq/RDA-landscape-genomics/tree/main).

Supplemental material available at GENETICS online.

## Acknowledgments

The authors are grateful to Beat Keller and Ueli Grossniklaus for providing space for the flowering-time experiments as well as the Amasino lab, Sam Yeaman, and Dieter Ebert for stimulating discussion on flowering time and polygenic selection. We also thank Ludmila Tyler for sharing seeds. We eventually would like to thank the two anonymous reviewers for their time and very constructive comments.

## Funding

We would like to thank the Swiss National Science Foundation (project 31003A_182785) and the Research Priority Program Evolution in action from the University of Zürich for their generous funding.

## Conflicts of interest

The authors declare no conflict of interest.

## Author contribution

N.M. conceived the study, performed the greenhouse experiment, contributed to the analysis and the writing of the manuscript. M.T. collected samples, contributed to the analysis. W.X., R.H., and L.K. contributed to the analysis. D.P.W. contributed to the writing of the manuscript. A.C.R. conceived the study, contributed to the analysis and the writing of the manuscript.

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

*Editor: A. Paterson*