## [Peer Review File · Genetics]

Polygenic architecture of flowering time and its relationship with local environments in the grass *Brachypodium distachyon*.

Nikolaos Minadakis, Lars Kaderli, Robert Horvath, Yann Bourgeois, Wenbo Xu, Michael Thieme, Daniel Woods, and Anne Roulin

NOTE: The reviews and decision letters are unedited and appear as submitted by the reviewers. In extremely rare instances and as determined by a Senior Editor or the EIC, portions of a review may be redacted. If a review is signed, the reviewer has agreed to no longer remain anonymous. The review history appears in chronological order.

Review Timeline:

Submission Date:	2023-08-23
Editorial Decision:	2023-10-10
Resubmission Received:	2024-01-12
Accepted:	2024-03-07

October 10, 2023

GENETICS-2023-306451

Polygenic architecture of flowering time and its relationship with local environments in the grass *Brachypodium distachyon*.

Dear Dr. Roulin:

Two experts in the field have reviewed your manuscript, and I have read it as well, finding much interest in it. While your manuscript is not currently acceptable for publication in GENETICS, we would welcome a substantially revised manuscript. Both reviewers have comments and concerns to be addressed in a revised manuscript. You can read their reviews at the end of this email.

We look forward to receiving your revised manuscript. Please let the editorial office know approximately how long you expect to need for revisions.

Upon resubmission, please include:

1. A clean version of your manuscript;
2. A marked version of your manuscript in which you highlight significant revisions carried out in response to the major points raised by the editor/reviewers (track changes is acceptable if preferred);
3. A detailed response to the editor's/reviewers' feedback and to the concerns listed above. Please reference line numbers in this response to aid the editor and reviewers.
4. Please identify a public repository from which all salient data are available.

Your paper will likely be sent back out for review.

Additionally, please ensure that your resubmission is formatted for GENETICS
<https://academic.oup.com/genetics/pages/general-instructions>

Follow this link to submit the revised manuscript: Link Not Available

Sincerely,

Andrew Paterson
Associate Editor
GENETICS

Approved by:
Mario Calus
Senior Editor
GENETICS

Reviewer #1 (Comments for the Authors (Required)):

This manuscript authored by Minadakis et al. presents a quantitative genetics study on flowering time variation in *Brachypodium distachyon* natural accessions by combining bioclimatic variables and trait measurements from both greenhouse and natural settings. In that sense it is a novel project that contributes to our understanding of the polygenic architecture of flowering time in grasses and offers valuable insights into its connection with environmental adaptation.

A few questions and concerns come to mind as I read the manuscript:

1. Line 193: what is the population structure of accessions used in this study? Is this population structure correlated with their geographical origins?
2. Line 244-250: How many accessions with outdoor flowering time measurements from Stritt et al. 2022 are used for this study? Given that there were 105 accessions measured in 2017, it would be helpful to clarify whether the accessions used in this study are considered as good representatives for all available lines. Additionally, the authors should present and interpret the results of flowering time in the accessions grown in greenhouse and outdoor in a more comparative way.

3. Line 251-258: After the filtering process, how many SNPs in total were retained for the association study? What is the general marker density across the genome?

My concern with this GWAS section is its statistical power to identify significant peaks. One year of phenotypic data for association mapping, especially for a polygenic trait, might be insufficient. Both the acquisition and utilization of the phenotypic (flowering time under outdoor conditions) and genetic data (SNPs) should be better described in the current study. This clarification is necessary because the lack of significant associations overlapping with known flowering-time genes (Figure 5D) cannot be solely attributed to environmental or gene-environment interactions. The authors should provide more evidence to support their claim that these flowering-time genes may not largely contribute to the flowering time variation in *Brachypodium distachyon* in the wild (line29). Multiple genes with relatively small to moderate effect sizes (undetected in this study) could collectively have more substantial additive effects on a polygenic trait.

Other minor comments:

Line 76: please provide more context regarding the term "near base-perfect reference genome".

Fig. 1B, 1C: please add the unit of precipitation.

Line 339: Table 1 only presents environmental variables, not genetic clades.

Line 353: spell out the full name of AFT-genes.

Line 604-629: consider a more concise presentation for this section, as the discussion of flowering-gene functions and regulatory mechanisms may not be directly relevant to the main topic of this study.

Line 733-734: delete one of the duplicate "therefore".

Line 743-744: please restructure this sentence.

Reviewer #2 (Comments for the Authors (Required)):

The manuscript by Minadakis et al. explores the evolutionary genetics of natural variation in flowering time and its response to vernalization across natural accessions of the genetic model plant *Brachypodium distachyon*. The work is a helpful addition to a small but growing and important literature implicating selection as an evolutionary force capable of maintaining long-range linkage disequilibrium. The authors thoughtfully analyze their data in light of the population structure inherent in their sampling, and they provide complementary analyses with appropriate genome-wide controls that evaluate alternate hypotheses for their observations. In addition, the manuscript also builds on a series of results over the last two decades or so finding that the loci detected as underlying genetic variation in a trait when observed in controlled or greenhouse conditions are not the same as those detected in field conditions relevant to natural populations. In general, I thought the work was well designed, described, and notable, and I have just a handful of major comments that I would ask the authors to consider as they revise their manuscript.

Quantifying flowering time parameters: Two measures of flowering time are extracted from the observed vernalization reaction norms. Minimum DTF with and minimum DTF without including the vernalization period. At least three other notable parameters could be extracted from the dataset using actual or modeled (Donohue et al. 2015 TREE) values: the minimum threshold duration of vernalization necessary to be permissive for flowering, the DTF when that minimum vernalization period is received, and the saturating threshold duration of vernalization above which no further reduction in flowering time is gained. Depending on how correlated these parameters with the flowering parameters already assessed, they may track different environmental variables, informing additional hypotheses about how local conditions act as agents of selection on phenology, and may help parse what GWAS signals relate to what aspects of the regulatory control of the floral transition. I encourage the authors to consider how best to infer additional parameters like these for parallel use in all their downstream analyses, and a sentence or two describing the developmental and/or ecological rationale for choosing the measures of DTF that they have chosen would improve interpretation by the reader as well.

Reporting of climate parameters: How spatially correlated are the environmental factors with each other? Could solar rad and precipitation be yielding mostly the same information? Please add a supplemental figure showing the pairwise correlations among variables and the climate PCA, and a supplemental table with the climate PC1 and PC2 loadings. The climate parameter data should also be addressed in the data availability statement.

Genotype-environment association analysis: A recurring theme is that population structure and flowering time variation covary, potentially leading to weaker magnitude or false negative results when population structure is controlled for. For the gene-environment association analysis, the extent of this problem could be quantified using RDA, which provides estimates of how much of the total genetic variation is explained by structure, environment, or both.

Additional information about field experiment: The authors provide convincing information that the historic average conditions at the Zürich field site are similar to those found in natural populations at the northern edge of the species range. Including some additional information about the specific conditions encountered and behavior of the plants during the experimental season

would be helpful for the reader to know as well. For instance, how did the seasonal weather conditions in Zurich during the year of the field experiment compare to average conditions? How was the phenology the plantings there consistent with or different from the trajectories of seedlings in other similar locations (that year or in other years)? Were any accessions that didn't not flower in the greenhouse included, and if so, did they flower outdoors?

Field Experiment GWAS hit: It is exciting that the hit found by the authors shows environmental association and a signature of selection for population differentiation like the AFT SNPs. Since the annotated function of the gene most proximate to the SNP does not point to anything immediately noteworthy, is there other information that can be mined for corroboration? For example, expression data showing differential expression during the correct developmental stage or in response to cold treatment? Or using the pan-genomic resources available in the species, are there any PAVs in more obvious candidates nearby that the SNP could be tagging?

Line 46 - why necessarily "favorable"?

Line 383: need plural - "accessions"

Line 626: spell out "long days" for "LD"

Line 629: substitute "among" for "in"

Line 648: substitute "extend" for "extent"

Supplemental Figures: Some of the captions refer to SFT rather than AFT genes.

Fig. S2: If readable, please include bars over the top to indicate which SNPs correspond to which loci.

Fig. S4: Please provide a legend for the colors.

Answers to Reviewer #1

This manuscript authored by Minadakis et al. presents a quantitative genetics study on flowering time variation in *Brachypodium distachyon* natural accessions by combining bioclimatic variables and trait measurements from both greenhouse and natural settings. In that sense it is a novel project that contributes to our understanding of the polygenic architecture of flowering time in grasses and offers valuable insights into its connection with environmental adaptation.

A few questions and concerns come to mind as I read the manuscript:

1. Line 193: what is the population structure of accessions used in this study? Is this population structure correlated with their geographical origins?

Indeed, there is a correlation between structure and geographical origin in *B. distachyon*. We do believe that this information is displayed by the maps and the legends (e.g. figure 1) "Map displaying the location of a given accession as well as their genetic clade of origin (C in yellow, A_East in magenta; A_Italy in turquoise; B_East in green and B_West in dark blue)."

2. Line 244-250: How many accessions with outdoor flowering time measurements from Stritt et al. 2022 are used for this study? Given that there were 105 accessions measured in 2017, it would be helpful to clarify whether the accessions used in this study are considered as good representatives for all available lines. Additionally, the authors should present and interpret the results of flowering time in the accessions grown in greenhouse and outdoor in a more comparative way.

31 accessions are common to the two experiments. This is due to the fact that many accessions have been collected since Stritt et al. to fill-up geographical gaps (eg. Balkans and southern Spain). We did not discover new genetic clades with the entire diversity panel (Minadakis et al 2023) indicating that the set of accessions we used for the outdoor experiment are good representative of the entire panel regarding genetic diversity. We have now added this information in the outdoor experiment section. We hope that the new version finally provides a better comparison between the two experiments.

3. Line 251-258: After the filtering process, how many SNPs in total were retained for the association study? What is the general marker density across the genome?

The filtering process resulted in 2,266,225 SNPs. We added the information to the text in the corresponding result section. Considering that the genome is small (272 Mb, information added in the introduction), the density of markers remains high. We added a supplementary figure as an illustration of this result (Figure S9)

4. My concern with this GWAS section is its statistical power to identify significant peaks. One year of phenotypic data for association mapping, especially for a polygenic trait, might be insufficient. Both the acquisition and utilization of the phenotypic (flowering time under outdoor conditions) and genetic data (SNPs) should be better described in the current study. This clarification is necessary because the lack of significant associations overlapping with known flowering-time genes (Figure 5D) cannot be solely attributed to environmental or gene-environment interactions. The authors should provide more evidence to support their claim that these flowering-time genes may not largely contribute to the flowering time variation in *Brachypodium distachyon* in the wild (line29). Multiple genes with relatively small to moderate effect sizes (undetected in this study) could collectively have more substantial additive effects on a polygenic trait.

We have largely re-written and toned-down this part, especially in the light of the redundancy analyses suggested by reviewer 2. We indeed over-interpreted our data.

In brief, we now conclude that the GWAs allow to identify potentially unknown genetic factors but do not allow to exclude the classical flowering-time genes (as initially claimed), as population structure clearly reduces our detection power. We however believe that the methods sections describe how the data were acquired and analyzed. As referred to in the text, the genetic data have been published last year and the scripts used for generating them made publicly available (Minadakis et al. 2023).

Other minor comments:

Line 76: please provide more context regarding the term "near base-perfect reference genome".

We change "near base-perfect reference genome" by "chromosome-level genome assembly"

Fig. 1B, 1C: please add the unit of precipitation.

The figure has been changed and this information is not displayed anymore.

Line 339: Table 1 only presents environmental variables, not genetic clades.

LMM analyses have been replaced by Redundancy analyses (RDA, see reviewer 2's comment).

Line 353: spell out the full name of AFT-genes.

This part has been changed as well. We provide the full names when we refer for the first time to flowering time genes.

Line 604-629: consider a more concise presentation for this section, as the discussion of flowering-gene functions and regulatory mechanisms may not be directly relevant to the main topic of this study.

We have drastically reduced this section.

Line 733-734: delete one of the duplicate "therefore".

We removed the second therefore.

Line 743-744: please restructure this sentence.

We have restructured this sentence.

Answers to Reviewer #2

The manuscript by Minadakis et al. explores the evolutionary genetics of natural variation in flowering time and its response to vernalization across natural accessions of the genetic model plant *Brachypodium distachyon*. The work is a helpful addition to a small but growing and important literature implicating selection as an evolutionary force capable of maintaining long-range linkage disequilibrium. The authors thoughtfully analyze their data in light of the population structure inherent in their sampling, and they provide complementary analyses with appropriate genome-wide controls that evaluate alternate hypotheses for their observations. In addition, the manuscript also builds on a series of results over the last two decades or so finding that the loci detected as underlying genetic variation in a trait when observed in controlled or greenhouse conditions are not the same as those detected in field conditions relevant to natural populations. In general, I thought the work was well designed, described, and notable, and I have just a handful of major comments that I would ask the authors to consider as they revise their manuscript.

Quantifying flowering time parameters: Two measures of flowering time are extracted from the observed vernalization reaction norms. Minimum DTF with and minimum DTF without including the vernalization period. At least three other notable parameters could be extracted from the dataset using actual or modeled (Donohue et al. 2015 TREE) values: the minimum threshold duration of vernalization necessary to be permissive for flowering, the DTF when that minimum vernalization period is received, and the saturating threshold duration of vernalization above which no further reduction in flowering time is gained. Depending on how correlated these parameters with the flowering parameters already assessed, they may track different environmental variables, informing additional hypotheses about how local conditions act as agents of selection on phenology, and may help parse what GWAS signals relate to what aspects of the regulatory control of the floral transition. I encourage the authors to consider how best to infer additional parameters like these for parallel use in all their downstream analyses, and a sentence or two describing the developmental and/or ecological rationale for choosing the measures of DTF that they have chosen would improve interpretation by the reader as well.

We thank the reviewer for this very constructive suggestion. We have now extracted from the greenhouse experiment three additional flowering time-related traits as suggested. Since not all accessions undergo harsh winter (see new Figure S2), we believe that including the minimum threshold duration of vernalization necessary to be permissive for flowering is indeed especially relevant, as now justified in the text. The four flowering time traits show significant but relatively low levels of correlation (updated figure 1). We have not included germination time as most plants germinated within 5 days (TableS1).

Reporting of climate parameters: How spatially correlated are the environmental factors with each other? Could solar rad and precipitation be yielding mostly the same information? Please add a supplemental figure showing the pairwise correlations among variables and the climate PCA, and a supplemental table with the climate PC1 and PC2 loadings. The climate parameter data should also be addressed in the data availability statement.

We have added a supplementary table the bioclimatic variable extracted at each site, as well as the PCA axes. Indeed, bioclimatic variables are correlated in our system as previously shown by Minadakis et al (2023). We now include a correlogram in figure S2.

We have added the climate parameters in the data availability statement.

Genotype-environment association analysis: A recurring theme is that population structure and flowering time variation covary, potentially leading to weaker magnitude or false negative results when population structure is controlled for. For the gene-environment association analysis, the extent of this problem could be quantified using RDA, which provides estimates of how much of the total genetic variation is explained by structure, environment, or both.

We have replaced the linear mixed model analyses by RDA to test for association between flowering time-related traits and bioclimatic variables or SNPs at flowering time genes. While we recover an association between flowering time-related traits and bioclimatic variables (speaking in favor of local adaptation), disentangling the confounding effect of population structure (genetic clade) and SNPs at flowering time genes remain challenging.

We believe that the high level of differentiation at flowering-time genes (F_{st} and X_{tx}) as well as the sign of co-selection (and molecular characterization done by other groups) speak in favor of a role in flowering-time genes. We have changed the results and discussion accordingly.

Additional information about field experiment: The authors provide convincing information that the historic average conditions at the Zürich field site are similar to those found in natural populations at the northern edge of the species range. Including some additional information about the specific conditions encountered and behavior of the plants during the experimental season would be helpful for the reader to know as well. For instance, how did the seasonal weather conditions in Zurich during the year of the field experiment compare to average conditions? How was the phenology the plantings there consistent with or different from the trajectories of seedlings in other similar locations (that year or in other years)? We have unfortunately little information to provide here as we did not monitor temperature during the experiment.

Were any accessions that didn't not flower in the greenhouse included, and if so, did they flower outdoors?

Four of the five accessions that did not flower in the greenhouse were included in the outdoor experiment. All four flowered. We added a sentence to clarify this.

Field Experiment GWAS hit: It is exciting that the hit found by the authors shows environmental association and a signature of selection for population differentiation like the AFT SNPs. Since the annotated function of the gene most proximate to the SNP does not point to anything immediately noteworthy, is there other information that can be mined for corroboration? For example, expression data showing differential expression during the correct developmental stage or in response to cold treatment? Or using the pan-genomic resources available in the species, are there any PAVs in more obvious candidates nearby that the SNP could be tagging?

Expression data are unfortunately not available for a large number of accessions in *B. distachyon* and certainly not in conditions that may be related to flowering induction. Regarding the pan-genome, most of the accessions used originate from the B_lineage (B_East and B_West) and do not represent all the diversity used in this study. We believe that validating T-DNA lines or applying CRISPR-cas9 technology would constitute a more direct approach to validate the gene function. This is out of the scope of the study but constitute an important goal for the future.

Line 46 - why necessarily "favorable"?

We removed favorable.

Line 383: need plural - "accessions"

Corrected

Line 626: spell out "long days" for "LD"

This part has been removed (see comments to reviewer 1)

Line 629: substitute "among" for "in"

Corrected

Line 648: substitute "extend" for "extent"

Corrected

Supplemental Figures: Some of the captions refer to SFT rather than AFT genes.

We removed this term throughout the manuscript as most flowering time genes show a significant association with flowering time when the cluster effect is not taken into account.

Fig. S2: If readable, please include bars over the top to indicate which SNPs correspond to which loci.

FigureS4 now display the start of each gene.

Fig. S4: Please provide a legend for the colors.

We added the legend for the colors

March 7, 2024

RE: GENETICS-2024-306789

Dr. Anne Roulin
University of Zurich
Plant and Microbial Biology
Zollikerstrasse 107
Zurich
Switzerland

Dear Dr. Roulin:

Congratulations! We are delighted to inform you that your manuscript entitled "**Polygenic architecture of flowering time and its relationship with local environments in the grass *Brachypodium distachyon*.**" is acceptable for publication in GENETICS. Many thanks for submitting your research to the journal.

To Proceed to Production:

1. Format your article according to GENETICS style, as discussed at <https://academic.oup.com/genetics/pages/general-instructions>, and upload your final files at <https://genetics.msubmit.net>.
2. Your manuscript will be published as-is (unedited-as submitted, reviewed, and accepted) at the GENETICS website as an Advanced Access article and deposited into PubMed shortly after receipt of source files and the completed license to publish. Please notify sourcefiles@thegsajournals.org if you do not wish to publish your article via Advanced Access.
3. We invite you to submit an original color figure related to your paper for consideration as cover art. Please email your submission to the editorial office or upload it with your final files. You can submit a small-sized image for evaluation, and if selected, the final image must be a TIFF file 2513px wide by 3263px high (8.375 by 10.875 inches; resolution of 600ppi). Please avoid graphs and small type.

If you have any questions or encounter any problems while uploading your accepted manuscript files, please email the editorial office at sourcefiles@thegsajournals.org.

Sincerely,

Andrew Paterson
Associate Editor
GENETICS

Approved by:
Mario Calus
Senior Editor
GENETICS

note: Please add jnls.author.support@oup.com and genetics.oup@kwglobal.com (or the domains @oup.com and @kwglobal.com) to your email program's "safe senders" list. You will be contacted by both at various points during the production process.

Review comments (if applicable):

Reviewer #1 (Comments for the Authors (Required)):

I appreciate the authors for addressing my concerns. I no longer have any other comments. Additional editorial suggestions may be necessary for improving the readership.